# Repeated and Interrupted Resistance Exercise Induces the Desensitization and Re-Sensitization of mTOR-Related Signaling in Human Skeletal Muscle Fibers

**DOI:** 10.3390/ijms23105431

**Published:** 2022-05-12

**Authors:** Daniel Jacko, Kirill Schaaf, Lukas Masur, Hannes Windoffer, Thorben Aussieker, Thorsten Schiffer, Jonas Zacher, Wilhelm Bloch, Sebastian Gehlert

**Affiliations:** 1Department of Molecular and Cellular Sports Medicine, Institute of Cardiovascular Research and Sports Medicine, German Sport University Cologne, 50933 Cologne, Germany; d.jacko@dshs-koeln.de (D.J.); k.schaaf@dshs-koeln.de (K.S.); lukas.masur@gmx.de (L.M.); h.windoffer@googlemail.com (H.W.); ausska@gmx.de (T.A.); w.bloch@dshs-koeln.de (W.B.); 2Olympic Base Center NRW/Rhineland, 50933 Cologne, Germany; 3Outpatient Clinic for Sports Traumatology and Public Health Consultation, German Sport University Cologne, 50933 Cologne, Germany; t.schiffer@dshs-koeln.de; 4Department ofPreventative and Rehabilitative Sports and Performance Medicine, Institute of Cardiology and Sports Medicine, German Sports University Cologne, 50933 Cologne, Germany; j.zacher@dshs-koeln.de; 5German Research Centre of Elite Sport (Momentum), German Sport University Cologne, 50933 Cologne, Germany; 6Institute of Sport Science, Biosciences of Sports, University of Hildesheim, 31141 Hildesheim, Germany

**Keywords:** anabolic signaling, mTOR, ribosomal protein S6, p70S6k, desensitization, hypertrophy, resistance exercise, type I myofibers, type II myofibers, skeletal muscle, unloading, desmin, HSPB5

## Abstract

The acute resistance exercise (RE)-induced phosphorylation of mTOR-related signaling proteins in skeletal muscle can be blunted after repeated RE. The time frame in which the phosphorylation (_p_) of mTOR^S2448^, p70S6k^T421/S424^, and rpS6^S235/236^ will be reduced during an RE training period in humans and whether progressive (PR) loading can counteract such a decline has not been described. (1) To enclose the time frame in which _p_mTOR^S2448^, _p_rpS6^S235/236^, and _p_p70S6k^T421/S424^ are acutely reduced after RE occurs during repeated RE. (2) To test whether PR will prevent that reduction compared to constant loading (CO) and (3) whether 10 days without RE may re-increase blunted signaling. Fourteen healthy males (24 ± 2.8 yrs.; 1.83 ± 0.1 cm; 79.3 ± 8.5 kg) were subjected to RE with either PR (*n* = 8) or CO (n = 6) loading. Subjects performed RE thrice per week, conducting three sets with 10–12 repetitions on a leg press and leg extension machine. Muscle biopsies were collected at rest (T0), 45 min after the first (T1), seventh (T7), 13th (T13), and 14th (X-T14) RE session. No differences were found between PR and CO for any parameter. Thus, the groups were combined, and the results show the merged values. _p_rpS6^S235/236^ and _p_p70s6k^T421/S424^ were increased at T1, but were already reduced at T7 and up to T13 compared to T1. Ten days without RE re-increased _p_rpS6^S235/236^ and _p_p70S6k^T421/S424^ at X-T14 to a level comparable to that of T1. _p_mTOR^S2448^ was increased from T1 to X-T14 and did not decline over the training period. Single-fiber immunohistochemistry revealed a reduction in _p_rpS6^S235/236^ in type I fibers from T1 to T13 and a re-increase at X-T14, which was more augmented in type II fibers at T13 (*p* < 0.05). The entity of myofibers revealed a high heterogeneity in the level of _p_rpS6^S235/236^, possibly reflecting individual contraction-induced stress during RE. The type I and II myofiber diameter increased from T0 and T1 to T13 and X-T14 (*p* < 0.05) _p_rpS6^S235/236^ and _p_p70s6k^T421/S424^ reflect RE-induced states of desensitization and re-sensitization in dependency on frequent loading by RE, but also by its cessation.

## 1. Introduction

Resistance exercise (RE) is the essential training mode for increasing strength and muscle mass. In 2020, for the first time, the WHO recommended the implementation of RE “two to three times per week with moderate or greater intensity” in one’s regular lifestyle to maintain health [1].

Muscle mass is finally gained when protein synthesis induced by RE increasingly exceeds its breakdown [2]. The initiation of protein synthesis in skeletal muscle is mainly controlled by the mTOR signaling pathway [3]. Amino acid abundance and mechanical and metabolic signals converge in this pathway, and downstream proteins, such as p70S6k and rpS6, are activated by phosphorylation (_p_) [4,5,6]. To date, both proteins have often been used to estimate the anabolic response of skeletal muscle following RE [7,8,9]. Importantly, depending on the tissue [9,10,11], the timing of biopsy sampling [12], the time frame of observation, and nutritional state [13], the phosphorylation of mTOR, p70S6k, and rpS6 does or does not correlate with protein synthesis [9,14,15,16], similarly to protein synthesis with hypertrophy [17,18]. Nonetheless, mTOR is a master activator of protein synthesis [7,19], and its inhibition via rapamycin shuts down protein synthesis in *human* skeletal muscle after RE [3].

While the recommendation of the WHO is highly appreciated, it raises the question of what the difference between moderate or greater intensity means for acute molecular muscle responses of mTOR-related signaling. It is also unclear whether RE repeated in the long term has to be continued without interruption, how acute molecular responses behave under repeated training, and whether interruptions of RE may change the signaling pattern in skeletal muscle.

To date, numerous studies have determined the increased phosphorylation of p70S6k and rpS6 after RE, showing that the intensity [12,20] and volume of RE [21,22] modulate this response in whole skeletal muscle lysates and single myofibers [12,23]. While many of these studies described the anabolic signaling acutely after RE, only a few determined the acute signaling response after a period of training. In rat and *human* skeletal muscle, the acute phosphorylation is blunted after weeks of training, which indicates that skeletal muscle is somehow desensitized as a result of repeated resistance training [15,24]. Because long-term training programs may benefit from maintaining the anabolic signaling response on a high level, it would be of great interest to determine the time frame in which signaling will start to be reduced when RE is frequently carried out.

In the same cohort of subjects as that used here, we recently showed that, under conditions of repeated RE, the accumulation of the z-disk protein filamin C increases in response to repeated RE [25,26]. This was associated with a reduction in myofibrillar damage [25], which is an important prerequisite for the accumulation of muscle proteins and hypertrophy [27]. We further showed in our study that the phosphorylation of HSPB5 at Serine 59 (_p_HSPB5^S59^), a small chaperone regulated by mechanical-stress-induced protein unfolding [28], is significantly reduced after repeated RE [26]. This response coincided with increased desmin abundance, an intermediate filament that is important for the structural and mechanical integrity of the contractile apparatus in muscle fibers [29]. After 10 days without RE, there was a significant re-increase in HSPB5^S59^ phosphorylation and a reduction in desmin, indicating that skeletal muscle sarcomeres also rapidly respond to phases of unloading [26]. Because this rapid adaptation towards repeated RE may also affect the phosphorylation of anabolic signaling proteins [24], in muscle samples of 14 male subjects who were subjected to either progressive (PR) or constant (CO) loading by repeated RE, we analyzed whether—and in which time frame—reduced _p_mTOR^S2448^, _p_p70S6k^T421/S424^, and _p_S6^S235/236^ in skeletal muscle may occur. We also analyzed whether cessation of RE for 10 days may reestablish this response, as reflected by a re-increased phosphorylation of mTOR, p70S6k, and rpS6. Because whole skeletal muscle signaling, as determined by Western blotting, does not highlight fiber-type-specific responses, we also analyzed whether a changed phosphorylation of rpS6 would be differently regulated between and within type I and II fibers [23]. We hypothesized that the phosphorylation of mTOR-related signaling proteins will be significantly reduced within several weeks of repeated RE and that PR will prevent that decrease. We further anticipated that individual myofibers will reflect this response. We also expected that ten days of unloading will already significantly augment the previously reduced signaling to the initial levels after the first session of training.

## 2. Results

In contrast to our hypothesis that significant differences between PR and CO would occur over the time course with repeated RE, we did not observe differences between PR and CO in terms of the acute phosphorylation of mTOR, p70S6k, or rpS6 or in terms of changes in diameter by fiber type (data not shown). Therefore, we merged all subjects into one single group (*n* = 14) when presenting the following datasets.

### 2.1. Western Blotting

The analysis of _p_rpS6^S235/236^ (effect size (ES): 0.521) showed a significant upregulation following the first bout of resistance exercise (T1) in the untrained state when compared to rest (T0) (*p* < 0.001) (Figure 1A). After seven RE sessions (T7) and up to T13, _p_rpS6^S235/236^ was only moderately increased compared to the resting conditions (T0; *p* < 0.05), but was significantly reduced compared to the initial response at T1 (*p* < 0.05). Interestingly, after the detraining phase and the 14th RE session (X-T14), _p_rpS6^S235/236^ was increased again, as it was not significantly different from T1, in contrast to T7 and T13. The regulation of _p_p70S6k^T421/S424^ (ES: 0.56) (Figure 1B) over the time course of our study showed basically the same pattern as that described for _p_rpS6^S235/236^, with the exception that the re-increase in _p_p70s6k^T421/S424^ after X-T14 was more moderate but with less heterogeneity. The total levels of p70S6k and rpS6 remained unchanged.

In contrast to that of _p_rpS6^S235/236^ and p70S6k^T421/S424^, the phosphorylation of _p_mTOR^S2448^ (ES: 0.316) was significantly increased at T1 (*p* < 0.05), T7 (*p* < 0.05), T13 (*p* < 0.01), and X-T14 (*p* < 0.001) compared to that at T0 (Figure 1C). The _p_mTOR^S2448^ levels tended to increase over more and more time. 

#### 2.1.1. Fiber-Type-Specific Analysis of _p_rpS6^S235/236^

As there was no difference between T7 and T13, and for the reasons of the efficiency of the very time-demanding single-fiber analysis of _p_rpS6^S235/236^, we excluded T7 and focused on the primary end points of the study (T0, T1, T13, and X-T14). The fiber-type-specific examination of _p_rpS6^S235/236^ via IHC (Figure 2B–D) showed a similar regulation pattern for type I and II fibers, as observed in Western blotting (Figure 2A–D). In type I fibers, the reduction of _p_rpS6^S235/236^ from T1 to T13 lacked significance, but showed a clear tendency (*p* = 0.072) (ES: 0.70). In addition, for type II fibers (ES: 0.59), there was a clear but not significant reduction in _p_rpS6^S235/236^ after 13 RE sessions. However, after detraining, X-T14 exerted a significant increase in _p_rpS6^S235/236^ compared to T13 in type II fibers (*p* < 0.05), emphasizing the efficacy of reduced mechanical stimulation by RE for re-sensitizing the potential for phosphorylation of rpS6.

Our RE training, which involved a mixture of concentric and eccentric contractions with ~75–80% of the 1RM, likely exerted a strong activation of all fiber types (Figure 2A). The staining showed a clearly visible abundance of _p_rpS6^S235/236^ in type I, IIA, and some IIX fibers. In general, type I fibers seemed to show a stronger signal than type II fibers (Figure 2D). At T0 (ES: 0.57), and especially after unaccustomed RE at T1 (ES: 0.50), type I fibers showed a stronger _p_rpS6^S235/236^ than in type II fibers (*p* < 0.01) and a trend toward a difference at T13 with *p* = 0.06 (Figure 2D) (ES: 0.37).

We also descriptively included all _p_rpS6^S235/236^-stained and measured type I and type II fibers from all subjects at the time points T0, T1, T13, and X-T14 in a violin plot that displays a combination of frequency analysis and the individual level of the sarcoplasmic staining intensity of _p_rpS6^S235/236^ in individual fibers (Figure 2C). 

From time point to time point, the majority of the fibers of a fiber type showed a similar change in the level of _p_rpS6^S235/236^ (Figure 2C). However, it can be clearly seen that a proportion of the strongly stained fibers at T1 were lost during detraining (compare T1 and T13 for type I and II fibers in Figure 2C). Interestingly, there were always fibers that showed a very low staining intensity (Figure 2A and the violin plots in Figure 2C) or a very high staining intensity. At least in type II fibers, an extended range of strongly stained fibers was observed (T13 for type II fibers in Figure 2C), which somehow seemed to compensate for the strong reduction of _p_rpS6^S235/236^ in the majority of the fibers. This illustration expands the picture obtained in the Western blotting analysis and emphasizes the high variability between fibers of one type.

#### 2.1.2. Type I and Type II Fiber Diameter

The evaluation of fiber-type-specific muscle growth (Figure 3) revealed a significant increase in the diameter of type I fibers (ES: 0.34) after 13 RE sessions (*p* < 0.05), which was preserved after 10 days of training cessation (*p* < 0.05) (Figure 3, left panel). A similar increase was detected in type II fibers (ES: 0.27) (Figure 3, right panel), and both fibers increased in diameter to a similar extent.

## 3. Discussion

We analyzed the phosphorylation of mTOR-related anabolic signaling proteins in *human* skeletal muscle in response to repeated RE and, moreover, a short period of unloading by refraining from RE. It was hypothesized that an acute bout of unaccustomed RE would initiate a strong phosphorylation of mTOR, p70S6k, and rpS6, that repeated RE training would regulate a significant decline in this phosphorylation somewhere within five weeks of RE and that 10 days without RE would re-increase this phosphorylation. We further investigated whether progression of RE intensity would prevent a drop in acute RE-induced phosphorylation compared to constant loading.

Surprisingly, we did not observe a difference concerning the degree of hypertrophy and phosphorylation of mTOR, rpS6, and p70S6k between PR und CO. However, we were likely underpowered for this approach, especially when considering that there was only a small difference in the training stimulation used in each group. Therefore, we included all subjects in one group, thus raising the power of our approach.

We were able to enclose the time frame in which acute RE-induced phosphorylation of _p_p70S6k^T421/S424^ and _p_rpS6^S235/236^ is blunted in *human* skeletal tissue and becomes desensitized towards anabolic signaling following RE-induced loading. Interestingly, only seven sessions of repeated RE, when conducted thrice per week, significantly reduced the phosphorylation of rpS6 and p70S6k 45 min after RE. Secondly, we were able to induce a re-increase in this phosphorylation after 10 days of RE cessation, indicating a renewed sensitivity of the anabolic responses in *human* skeletal muscle towards RE. Thirdly, we determined that individual type I and type II fibers reflected the responses obtained in whole-tissue lysates and that both fiber types were similarly affected by a reduction in _p_rpS6^S235/236^. In this context, a considerable variability concerning individual fiber responses was observed, indicating that Western blotting alone may hide some information about a fiber-type-specific modulation of anabolic signaling in skeletal muscle.

The activation of mTOR-related signaling triggers translation initiation [10] and indicates the onset of protein synthesis in skeletal muscle fibers [9,11]. Therefore, downstream of mTOR, p70S6k and rps6 are often used to estimate muscle anabolism in *human* skeletal muscle [7,8,18,20,22]. When rapamycin is administered to *human* subjects after RE, mTOR activity is shut down and protein synthesis is blunted, which emphasizes the essential role of mTOR signaling in regulating protein synthesis [3]. However, only a few studies have analyzed the impact of repeated RE on acute phosphorylation of components of the mTOR cascade without repeated biopsies in *human* skeletal muscle within this time frame [15,30,31]. Our study design in *humans* is very closely related to an excellent animal study, and our results broadly corroborate their findings [24]. The authors determined that, as early as after 12 repeated sessions of electrical stimulation of rat gastrocnemius muscle, the phosphorylation of rpS6, p70S6k, and p90S6k was significantly reduced, but recovered after 12 days of unloading [24]. 

Changes in acute RE-induced signaling were also analyzed in other *human* studies. Wilkinson and colleagues observed a decreased phosphorylation of mTOR and rpS6 four hours after acute RE was conducted after 10 weeks of continued RT (thrice weekly) when compared to acute RE in an untrained state [30]. However, myofibrillar protein synthesis was still increased at 4 h after RE in the trained state and increased more than in the untrained state. In contrast, after 8 weeks of continued RE conducted twice per week with 70% and 90% of the 1RM, Figueiredo and colleagues determined that there was no reduction in _p_p70S6k^T389^ and _p_rpS6^S235/236^ signaling at 60 min post-RE [31]. Bamman and colleagues [15] found that, 24 h after acute RE—before and after a 16-week resistance training intervention—there was no reduction in the phosphorylation of rpS6 and p70S6k in either in older or younger subjects. In this study, the acute protein synthesis was increased in the young subjects but not in the old subjects, while an increase in fiber size was observed in both groups.

Obviously, there is still no consistent picture concerning the acute phosphorylation responses of mTOR-related signaling proteins during and after prolonged resistance training, and the training frequency during the intervention might account for that variability.

Indeed, our study corroborates only some of the results of acute phosphorylation responses in these *human* studies. As was already introduced, depending on the time point of biopsy sampling, and the training and nutritional states, the phosphorylation of mTOR, p70S6k, and rpS6 may or may not correlate with muscle hypertrophy or acute RE-induced protein synthesis [16]. This is especially because protein synthesis is increased from 4 to 72 h after RE [21,32,33,34], a timeframe in which the phosphorylation of mTOR, p70S6k, and rpS6 is already downregulated. We are, thus, fully aware that our responses of increased and decreased signaling via p70S6k and rpS6 cannot be directly related to increased protein synthesis in our samples. 

Enhanced ribosome biogenesis after repeated RE is one further mechanism by which an increase in translational efficiency can increase protein synthesis [35] in response to a bout of RE, or it can increase translational capacity independently of rpS6 or p70S6k phosphorylation [31]. 

The question is still the following: What drives the rapid decline in the phosphorylation levels after repeated RE, and how does it relate to muscle adaptation?

This study and previous ones have finally shown that hypertrophy is detectable when the acute phosphorylation of p70S6k and rpS6 is reduced. One possible explanation for the reduced phosphorylation of p70S6k and rpS6 is the reduced mechanical strain on the structural level of myofibers, which is believed to activate mTOR signaling via mechanisms that are still under discussion [36,37,38]. The generation of phosphatidic acids (PAs) from phospholipase D, which is located at the z-band in skeletal muscle, was proposed as one mechanism that activates mTOR [38,39]. It has been shown that mechanical stretching enriches PAs at the macropinosome and recruits mTOR to these sites, which increases downstream signaling [36]. Unfortunately, the detection of PA products was methodologically beyond the scope of our study, and therefore, the contribution of this mechanism in our approach remains speculative. It also does not fit our observations, as we did not detect reduced pmTOR^Ser2448^, but instead, we found constantly increased levels. Because _p_mTOR^S2448^ significantly regulates the activity of the mTOR complex, it contradicts a general reduction in the overall mTOR activity in our study. Therefore, our findings point to a discordance between increased _p_mTOR^S2448^ and the downstream proteins p70S6k and rpS6. However, it has to be considered that the only time point where we analyzed the phosphorylation of mTOR was 45 min after RE. We and others have shown that the acute RE-induced phosphorylation of mTOR, p70s6k, and rpS6 is blunted within a few hours after RE [12,17,21]. To safely detect the increased phosphorylation levels of mTOR-related proteins, we decided to collect biopsies at this time point and, therefore, much earlier than in the aforementioned studies. Indeed, the early time point enabled us to determine the acute phosphorylation levels under similar conditions, but did not inform us about the activity of proteins at 24–48 h after RE or between RE sessions. Hence, the general mTOR activity might still be blunted in the aftermath and after repeated RE sessions while being acutely phosphorylated.

However, it has also been shown that the mechanically induced activation of focal adhesion kinase (FAK), which is localized at skeletal muscle costameres, contributes to p70S6k phosphorylation, hence circumventing the direct regulation through mTOR [40]. Further, mitogen-activated protein kinases, especially the c-Jun-N-terminal kinase (JNK), have been shown to be highly sensitive markers for mechanical strain in skeletal muscle, and it has been shown that the phosphorylation of JNK increases with higher mechanical strain [12,41,42]. Additionally, JNK has been shown to increase the phosphorylation of p70S6k^T421/S424^ in overloaded skeletal muscle [43]. Using these very samples, in a recent study that determined the regulation of the small heat-shock protein member HSPB5, we showed that pJNK^T183/Y185^ was significantly reduced at T13 compared to T1 [26]. This indicates that repeated mechanical strain induced by RE somehow modifies the cellular environment of skeletal muscle fibers in a way in which mechanical sensitivity is reduced, anabolic signaling is affected, and p70S6k^T421/S424^ is reduced, as was partially explained in the present study. Indeed, MAPK kinase signaling has been shown to stimulate the phosphorylation of p90S6k, which can directly phosphorylate rpS6 [44,45] and circumvent the upstream activation by mTOR. In a study by Ogasawara, p90s6k was significantly reduced after 12 days of stimulation in rat skeletal muscle, indicating that this pathway could contribute to reduced _p_rpS6^S235/236^ [24].

It may be speculated that reduced mechanical straining of cytoskeletal components may have contributed to our observation, e.g., through RE-induced reinforcement of the cytoskeleton, which possibly dilutes the mechanical strain on cytoskeletal components of the sarcomere [46,47]. This is supported by the results of our past study, which was performed in the same cohort of subjects as that used here. There, we showed that the intermediate filament desmin was already significantly increased after seven RE sessions, probably leading to a reinforcement of the sarcomeric cytoskeleton and mediating an increase in overall sarcomeric stability [26]. This was supported by a simultaneous reduction in the phosphorylation of HSPB5, a small chaperone that is sensitive to unfolding of and damage to the sarcomeric protein [48]. Thus, it can be speculated that a reduction in mechanically induced _p_mTOR^S2448^ alongside or mediated by reduced deformation of components of the dystrophin–glycoprotein or integrin complex [49] may downregulate FAK and JNK activation and contribute to the blunted phosphorylation of p70S6k and rpS6, as observed in the current study.

Due to the early time frame of biopsy sampling, we also have limited insight into further molecular events that are triggered by acute RE and that persist for up to several days after loading. Next to protein synthesis, this involves muscle damage and mechanisms that regulate protein homeostasis [12,50]. Such events can contribute to adaptive events, which may explain the reduction in _p_rpS6^S235/236^ and _p_p70S6k^T421/S424^ to some degree.

Mechanical stimulation of skeletal muscle by RE induces myofibrillar damage and cytoskeletal disorganization in the z-disk at the beginning of an RT period when muscle is not adapted to increased mechanical stimulation [25,27,51,52]. Structural damage of muscles is commonly associated with the increased occurrence of creatine kinase (CK) or lactate dehydrogenase (LDH) in them [53]. Such damage is significantly reduced after repeated training [25,27], coinciding with a reduction in serum CK levels, as was also observed in the samples in our study [25]. We recently determined that RE-induced mechanical strain activates chaperone-assisted selective autophagy (CASA), which operates in the sarcomere [54]. This mechanism degrades but also synthesizes and accumulates filamin C (FLNc) in and around the z-disk of skeletal muscle [55], strengthens the sarcomere, and contributes to mechanoprotection [25,51]. This response was accompanied by a significant reduction in the chaperone response, as indicated by the blunted phosphorylation of HSPB5, which was shown in a separate study, but in the very same cohort of subjects [26]. The latter event indicated a significant change in the mechanosensitive environment of the myofibers, which was likely caused by the enhanced sarcomeric stability after repeated RE and the reduced necessity for augmented chaperone activity [56]. Interestingly, the desmin levels tended to decline after just 10 days of unloading, with a concomitant re-increase in HSPB5 phosphorylation in our recent study. In sum, the time frame of desmin accumulation and decreased HSPB5 phosphorylation [26], along with increased activation of CASA and accumulation of FLNC [25], coincides remarkably with the modulation of _p_rpS6^S235/236^ and _p_p70S6k^T421/S424^ phosphorylation in the current study.

Previously, we also showed that CASA is essentially mediated by the WW domain of BAG3, which recruits the tuberous–sclerosis complex (TSC) towards sites of mechanical damage within cells and the sarcomere [57]. This leads to inhibition of mTOR signaling at sites where autophagic degradation of damaged proteins is necessary. In contrast, in the remaining and undamaged environment of the sarcomere, TSC-mediated inhibition of mTOR is relieved and the anabolic response is increased, as indicated by the increased _p_rpS6^S235/236^ [57]. In this study, we also found significantly reduced muscle damage in our subjects after seven (T7) and 13 RE sessions (T13) [25]. Based on this mechanism, the reduced damage may have contributed to the observed reductions in _p_rps6^S235/236^ and _p_p70S6k^T421/S424^ after repeated training (T7 and T13) because, corresponding to this model, a higher number of free TSC complexes may inhibit mTOR throughout the fiber when they are not bound to mTOR complexes at sites of damage. This is much more likely at T1, when mechanical stress and myofibrillar damage are the highest [25]. The renewed increase in previously blunted _p_rpS6^S235/236^ after 10 days of unloading also fits into this picture well, as we found not only decreased desmin levels, but also a re-increased phosphorylation of HSPB5 [26]. Assuming that the phosphorylation of rpS6 and p70S6k correlates strictly with translational responses, according to our data, protein synthesis would be significantly reduced under conditions of structurally adapting myofibers. As discussed previously, this must not necessarily be the case. In contradiction to this assumption stands the observation that increased hypertrophy rather occurs when sarcomeric damage is reduced [27]. Further, due to the increased ribosomal efficiency in combination with reduced protein degradation after repeated RE [2], blunted _p_rpS6^S235/236^ may primarily reflect a structural adaptation of the sarcomeric environment. Under the aforementioned conditions, protein synthesis may, nevertheless, still induce a net increase in myofibrillar protein and muscle growth when damage (Figure 3) is reduced [25,27].

It also has to be considered that, despite the mTOR-mediated cap-dependent translation [58], the internal ribosome entry site (IRES)-mediated translation initiation occurs independently of the 7-methyl guanosine cap at the 5′ end of the mRNA [59]. In this way, the synthesis of muscle-specific proteins can be uncoupled from mTOR-dependent processes. IRES is obviously of crucial importance for muscle, as it was detected in utrophin-A mRNA [60]. Utrophin-A acts as a promising surrogate for missing dystrophin proteins in the muscles of patients with Duchenne muscular dystrophy (DMD) [60]. It is still unclear how repeated stimulation of *human* skeletal muscle and reduced _p_rpS6S^S235/236^ may change the proportion of IRES when mTOR and rpS6 activity is reduced. However, a recent paper determined that two months of aerobic exercise changed protein homeostasis in *human* skeletal muscle on the level of transcription, translation, and protein stability via chaperone-dependent mechanisms [61]. The authors also determined that IRES accounts for up to 33% of proteins associated with translation, protein degradation, and chaperone activity, all of which are mechanisms that are crucial for muscle adaptation.

We observed that both fiber types increased their diameter and showed significant changes in the phosphorylation of rpS6, which has been shown to occur in different training modes and time frames after RE [12,15,23]. Eccentric exercise [12,20] and intense RE with higher forces [12] specifically augment type II fiber responses, while intermediate programs [62], such as that used here, affect both fiber types. The response of _p_rpS6S^S235/236^ in IHC reflected the results obtained through Western blotting (Figure 1 and Figure 2) and confirmed the clear reduction in phosphorylation after T7 and T13 compared to the baseline; this result was also found within fiber types. However, the analysis of the entire count of _p_rpS6S^S235/236^-stained myofibers showed that they do not uniformly respond to desensitization (T1–T13) and re-sensitization (T13–X-T14) after repeated loading and unloading. Changes in myofiber recruitment may also account for this response, and a short-term immobilization changes the motoneuron recruitment within fiber types [63]. Despite the broad and uniform response of a great proportion of myofibers (Figure 2C), a smaller fraction showed substantial differences with a very high or a very low staining intensity at every time point. Based on the proposed mechanism of mechanical stimulation of mTOR signaling, the recruitment of those fibers during RE was either very high or relatively low. We and others showed that the RE-induced phosphorylation of HSPB5 and rpS6 is a fiber-type-specific event that depends on the contraction mode and training intensity [12,26] and that the recruitment of fibers may change under these conditions [52,64,65]. Repeated recruitment during RE may augment their structural adaptation and then reduce mechanical strain, leading to reduced signaling from T1 to T7 and T13. On the other hand, fibers that still display a high phosphorylation of rpS6 at T7 and T13 may not have previously adapted, but are highly recruited at these time points. Unfortunately, this remains speculative, as we cannot show data that could support these considerations.

We acknowledge some limiting factors of the current work. First, our study was well designed for investigating acute signaling responses under defined RE conditions, but it was likely too short and underpowered to determine potential differences between PR and CO, which may be detectable only after a number of months. Due to the restricted number of biopsies that could be collected in such a study, we focused on a post-loading time point (45 min) at which the phosphorylation was reported to be highly regulated [12,31,50]. However, we have no information about the phosphorylation and activity of those targets from either beyond this time point or between two RE sessions. Therefore, our argumentation of an mTOR-independent inhibition of _p_p70S6k^T421/S424^ and _p_rpS6^S235/236^, as well as a damage-dependent inhibition of mTOR via TSC proteins, remains descriptive and, to some degree, speculative. Finally, although we found clear differences between the fiber types, we have no data, e.g., EMG data, that could support potential changes in fiber recruitment during repeated RE and detraining in our study. Therefore, it remains unclear whether differential recruitment of myofibers over time or the adaptation of the structural environment of the fiber itself is responsible for altered signaling responses in the analyzed population of myofibers. 

Repeated resistance exercise significantly reduces the phosphorylation of anabolic signaling proteins with RE conducted thrice per week. This can be recovered by adding a 10-day microcycle without RE, indicating rapid structural modulations of the muscle fiber environment. Hence, our study reveals rapid changes in the phosphorylation of anabolic signaling proteins in skeletal muscle in response to repeated loading by RE in addition to unloading. Such a response likely reflects the adaptation and de-adaptation of the mechanosensitive environment within *human* skeletal muscle. Due to previous findings that were made in the same cohort of subjects, we were able to plausibly argue our assumptions and underline the potential causes of the effects that we observed here.

We highlighted that, up to the single-fiber level, the skeletal muscle environment is highly sensitive to the modulation of mechanical loading by RE and that the acute phosphorylation of p70s6k and rpS6 mirrors the adaptive process of skeletal muscle to some degree. To our knowledge, we are the first to show that anabolic signaling response in *human* skeletal muscle can be recovered by adding short microcycles without RE.

Timely coordinated reductions in training load within periodization approaches may reestablish the sensitivity of mTOR-related targets, but may also indicate the beginning of a de-adaptation of sarcomeric subsystems. Finally, it must be critically questioned whether a re-sensitization of the studied signaling actually represents a kind of refreshment or booster of the anabolic response and has a beneficial effect on muscle hypertrophy, or whether it is merely an expression of newly increased structural damage without a net positive anabolic effect. The latter would underpin the importance for training continuity in the maintenance of sarcomeric adaptations.

## 4. Methods

### 4.1. Ethics Statement

All protocols used in the present study align with the Declaration of Helsinki and were approved by the ethics committee of the German Sport University of Cologne (Approval Code: 10/2013). All subjects were informed orally as well as in writing about the purpose of the study and possible risks, and they gave written informed consent prior to participation.

### 4.2. Subjects

Subjects engaged in regular sports activities and had experience with resistance training, but were not specifically resistance trained (RE less than 2 times per week). All subjects were instructed to refrain from lower-limb resistance exercise for four weeks prior to the start of the study. Fourteen healthy male subjects (24 ± 2.8 yrs.; 1.83 ± 0.1 cm; 79.3 ± 8.5 kg) finally participated in the study. In previous studies (Gehlert et al. (2015)), we determined strong effects concerning the acute phosphorylation of rpS6 and p70S6k, which is why we also assumed a similar effect size in the current approach. Our sample size was justified by an a priori power analysis in G*power using an effect size (ES) of f = 0.30, alpha of 0.05, and power of 0.80. The analysis determined that 10 subjects were required for participation and for the later analysis of the time-dependent regulation of signaling proteins (within interactions). Retrospectively, no clear effects between groups (PR and CO) were determined, so the post hoc analysis determined that a minimum of 42 subjects per group would have been necessary. We acknowledge in the section on the limitations that we were underpowered for this approach.

### 4.3. Training Intervention

A detailed description of the study protocol can be found elsewhere [26]. Our primary aim was to apply sufficient mechanical stress to induce a significant increase in the phosphorylation of anabolic signaling proteins. Our intervention mirrored a classical resistance exercise training regimen reflecting a mesocycle of five weeks followed by a detraining phase of 10 days (Figure 4). Subjects performed 13 RE sessions within five weeks (3 times per week). After the 13th session, the detraining was applied, followed by a 14th and final RE session. Before RE, subjects cycled for 5 min on an ergometer with 1W/kg body weight. Thereafter, subjects performed a device-specific warmup (one set of 10 repetitions with 70% of the 10-repetition maximum training load). RE sessions consisted of 3 sets on a leg extension machine followed by 3 sets on a leg press machine (Gym80, Gelsenkirchen, Germany) with a resting phase of 120 s between sets and 180 s between exercises. Both exercises have been shown to sufficiently activate the vastus lateralis muscle [66]. RE requires the standardization of several parameters (e.g., loading, movement speed, and exercise mode) that can affect force generation and muscle activities [67].

Therefore, each contraction was carried out with a standardized cadence of two seconds in the concentric phase, two seconds in the eccentric phase, and one second isometric contraction phase during the change in movement. Notably, a recent work suggested that positive (concentric) and negative (eccentric) work can be a more suitable differentiation between those contraction modes [68]. To standardize the contraction pattern and time under tension, a strength-machine-coupled biofeedback system (DigiMax Messtechnik, Hamm, Germany) equipped with a WayCon SX80 movement sensor (WayCon Sx 80, Munich, Germany; precision in linearity: 0.02%) was used. The system was arranged in a way such that the subjects had to move a ball projected on a computer screen within the borders of a virtual tunnel in the defined time frame during each contraction.

During the RE, subjects trained at their 10-repetition maximum, which was determined on the respective training machines seven days prior the study. Subjects trained with either progressive loading (PR; *n* = 8), receiving 5% more weight loading when more than 12 repetitions were possible in the last set, or with constant loading (CO, *n* = 6), with an unchanged training weight over the study period and with repetitions not exceeding 12 per set. 

### 4.4. Tissue Collection and Analysis

Skeletal muscle biopsies from the *musculus vastus lateralis* were taken via the percutaneous needle biopsy technique [69] while at rest (T0) 7 days before the first exercise, as well as 45 min after the 1st (T1), 7th (T7), 13th (T13), and the 14th (X-T14) sessions. The biopsies were taken alternately from the left and right leg. Each biopsy on the same leg was taken in close proximity to the former one (~2–3 cm proximal or distal to the prior one), so all biopsies from one leg were finally collected within a range of 2–4 cm. After extraction, the muscle tissue was separated. The tissue samples used for Western blotting analysis were directly snap frozen in liquid nitrogen, and for immunohistochemistry samples, they were snap frozen in isopentane pre-cooled with liquid nitrogen. All samples were later stored at −80 °C until further processing. On the day of the muscle biopsies, RE was conducted between 08:00am and 09:00am. In preparation for the muscle biopsies, subjects fasted overnight, but ingested a dietary supplement shake (Fresubin energy drink; Fresenius Cabi, Germany) containing 20 g of protein, 24.8 g of carbohydrates and 13.4 g fat, 1260 kJ in total) as standardized meal one hour before RE. Water intake was allowed *ad libitum*.

### 4.5. Tissue Analysis

#### 4.5.1. Antibodies Used for Western Blotting

Primary: S6 ribosomal protein, total (*rabbit* monoclonal IgG AB; #4858; 1:1000); S6 ribosomal protein, phospho serin 235/236 (*rabbit* monoclonal IgG AB; #4858; 1:1000); p70S6 kinase, total (*rabbit* monoclonal IgG AB; #2708; 1:1000); p70S6 kinase, phospho threonin 421/serin 424 (*rabbit* polyclonal IgG AB; #9204; 1:1000); mTOR, phospho serin 2448 (*rabbit* polyclonal IgG AB; #5536; 1:1000); all previous ABs were from Cell Signaling Technology, Danvers, MA, USA; mTOR, total (*rabbit* polyclonal IgG AB; # ab137341; 1:1000; Abcam, Cambridge, UK); secondary: anti-*mouse* and anti-*rabbit* IgG, HRP-linked (# 7076 and # 7074; 1:7500; Cell Signaling Technology, Danvers, MA, USA).

#### 4.5.2. Antibodies Used for Immunohistochemistry

Primary: S6 ribosomal protein, phospho serin 235/236 (*rabbit* monoclonal IgG AB; #4858; 1:100; Cell Signaling Technology, USA); MyHCI and MyHCIIX (*mouse* monoclonal IgG AB; #A4.951 and *mouse* monoclonal IgM 6H1; 1:400 and 1:75; Developmental Studies Hybridoma Bank, Iowa City, IA, USA). Secondary: goat anti-*mouse* and goat anti-*rabbit*, polyclonal, biotinylated (#E0433 and #E0432; 1:400; Dako, Denmark).

#### 4.5.3. Western Blotting

After mechanical homogenization, muscle tissue was lysed in a triton-X100-based buffer (Cell Signaling, Beverly, MA, USA) in combination with a protease and phosphatase inhibitor cocktail (#78429, Thermo Scientific, Waltham, MA, USA). Finally, samples were spun at 13,600 rpm at 4 °C for 20 min, and supernatant was collected. Protein concentration was determined via a Lowry test kit (BioRad Laboratories GmbH, Munich, Germany). Homogenates of each subject were diluted to a protein concentration of 1.5 μg/μL of homogenate. Homogenates for analysis were thawed on ice, suspended in a 3x Laemmli buffer (0.5M Tris–HCl, 10% glycerol, 2% sodium dodecyl sulphate, 5% 2-mercaptoethanol, and 0.05% bromophenol blue) and heated at 95 °C for 7 min. Equal amounts of proteins (12 μg) for each subject and time point were separated on a 26-well, 4–12% BIS-TRIS Gel using a gel-casting system that worked with MOPS electrophoresis buffer. Buffers and equipment were all from BioRad (BioRad Laboratories GmbH, Munich, Germany). After electrophoresis, the gel was transferred to a polyvinylidene difluoride (PVDF) membrane (GE Healthcare Life Science, Amersham, UK) membrane by semidry blotting (Trans Blot Turbo, Bio-Rad, Hercules, CA, USA) for 34 min (1.2 A, 25 Vmax). Equal sample loading and transfer were checked by staining the gel with Coomassie and the PVDF membrane with Ponceau S after transfer.

Membranes were blocked for one hour at room temperature (RT) in 5% nonfat dry milk and dissolved in tris-buffered saline supplemented with 0.1% Tween20 (TBST), before being incubated with primary antibodies overnight at 4 °C. Then, membranes were washed, subsequently incubated with secondary antibodies, and diluted in TBST containing 5% nonfat dry milk for one hour at RT. After washing, membranes were incubated for 3 min with an enhanced chemiluminescence assay (ECL-Kit, GE Healthcare Life Science, Amersham, UK) and automatically captured (ChemiDoc MP, Bio-Rad, Hercules, CA, USA). Band densities were assessed semi-quantitatively using the ImageJ software (v. 1.53j; National Institute of Health, New York, NY, USA).

#### 4.5.4. Immunohistochemistry

Frozen muscle tissue was cut in 7-µm cross-sectional slices using a Leica CM 350 S cryo-microtome (Leica Microsystems, Nußbach, Germany) and placed on polysine microscope slides (VWR International, Leuven, Belgium). Muscle cross-sections from all biopsy time points of the subjects and groups were stained within a single batch while using the same antibody dilution and development time to minimize variability in staining efficiency. After air drying for one hour at RT, slides were placed in −20 °C acetone for eight minutes and air dried again before blocking for one hour at RT with 5% bovine serum albumin (BSA; fraction V) dissolved in 0.05 mM TBS. Then, samples were incubated with primary antibodies diluted in 0.05 mM TBS buffer (_p_S6^S235/236^ (1:100) and 6H1 (1:75)) overnight at 4 °C and subsequently washed before being incubated with the biotinylated secondary AB for one hour at RT (goat-anti *mouse* polyclonal biotinylated secondary antibodies (Dako Cytomation, Glostrup, Denmark) diluted to 1:400 in TBS. Slides were then incubated for one hour with streptavidin biotinylated horseradish peroxidase complex (Amersham Biosciences, Uppsala, Sweden) diluted to 1:400 in TBS and subsequently washed 3 times for 10 min in TBS. _p_rpS6^S235/236^ staining was finalized by a 3,3′ diaminobenzidine (DAB) solution. For determination of the fiber type, consecutively cut sections were stained as described above on separate slides. Type I fibers were identified with A4.951 antibodies and stained with DAB. Subsequently, type IIX fibers were identified with 6H1 antibody and stained with Fuchsin Red (K0625 Dako, Denmark). Type IIA fibers remained unstained. To confirm antibody specificity, control sections were incubated in TBS containing 0.8% BSA, but without primary antibodies. After dehydration, the stained sections were embedded in Entellan (Merck, Darmstadt, Germany) and supplied with a coverslip. 

#### 4.5.5. Quantification of Sarcoplasmic _p_S6^S235/236^ Staining

Between 8 and 12 digital photos of each time point and subject were taken from _p_rpS6^S235/236^-stained cross-sections at 10-fold magnification via a light microscope (KS-300, Zeiss, Germany) coupled with a digital CCD camera (Sony, Tokyo, Japan). The target-specific staining intensity for each myofiber was quantified through the selection of the sarcoplasmic region of the myofiber and its subsequent measurement through optical densitometry using the ImageJ software (v. 1.53j; National Institute of Health, USA). The details of the method are described elsewhere [12]. For the fiber-type-specific analysis of _p_rpS6^S235/236^, consecutively cut samples were analyzed; they were stained for type I and IIX fibers and then compared with the corresponding cross-section stained for _p_rpS6^S235/236^. On average, 60 ± 18 type I, 60 ± 17 type IIA, and 8 ± 9 type IIX fibers were determined per time point and subject. Because IIX fibers became less abundant during the training period [70], and in some subjects, they were not detectable at all, we refer in our manuscript to type I and type IIA fibers and exclude IIX fibers from our analysis. In sum, 3137 type I and 3094 type IIA fibers were used for the analysis of the entire fiber population.

### 4.6. Determination of Myofiber Diameter

Five to seven digital photos of each cross-section were captured in 20-fold magnification via a Zeiss KS-300 light microscope equipped with a digital CCD camera (Sony, Japan). By applying the specific pixel/aspect ratio of the 20X objectives used (2.4 pixels per µm), the best-fitting ellipse tool was applied using the ImageJ^®^ software (National Institutes of Health, USA) to determine the inner borders of the selected myofiber as the minor axis. At least 35 myofibers per fiber type (type I and type IIA), time point, and subject were analyzed for myofiber diameter. As type IIX fibers could not be found in every subject and time point, they were excluded from the myofiber diameter analysis.

### 4.7. Statistics

To check for the requirements of parametric tests (mixed ANOVA), normal distribution was assessed for each time point and target with the Kolmogorov–Smirmov test, and homogeneity of variances was determined with the Levene test. Nonparametric statistical tests were performed either due to inconsistencies in normal distribution (e.g., one time point was normally distributed and the other was not) or because uneven variances were given. To examine the differences across groups, a Kruskal–Wallis one-way ANOVA for independent samples was used. To test for differences in the responses between time points (T0 to X-T14), Friedman’s ANOVA for related samples was applied. Due to multiple comparisons, a Benjamini–Hochberg procedure was performed. Differences between fiber types were assessed by conducting the Wilcoxon signed-rank test. Data were normalized to T0 through the division of T0 to X-T14 by the mean value of T0. All statistics were carried out using the IBM SPSS software (version 25), and graphs were created with the GraphPad Prism software (version 6.01). The significance level was set to *p* < 0.05. Effect sizes are presented as Kendall’s W for parameters that were analyzed with Friedmann’s ANOVA and as “r” for the Wilcoxon comparison of _p_rpS6^S235/236^ between type I and II fibers.

It is of note that, because there were no statistical differences in the examined parameters between the two groups (PR and CO), we combined the subjects in one group, as was already done in a previous publication [26]. To comprehend group-specific and individual responses, individual data points were separated into group-specific colors (Figure 1, Figure 2 and Figure 3).

## Figures and Tables

**Figure 1 ijms-23-05431-f001:**
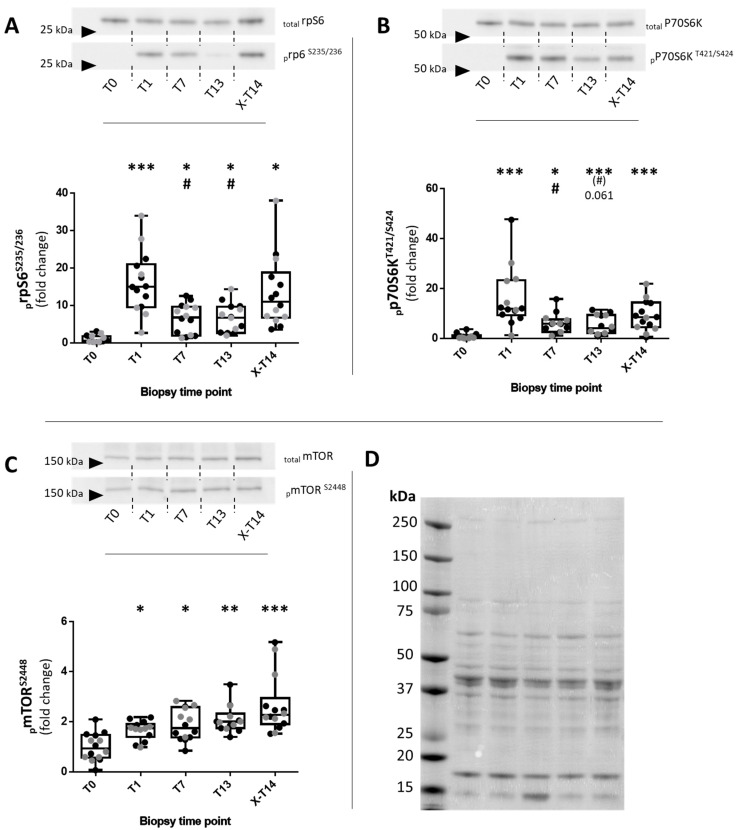
Western blotting analysis of acute responses of phosphorylated and total levels of mTOR, p70S6k, and rpS6 in *human* skeletal muscle during repeated RE. Regulation of _p_rpS6^S235/236^ (**A**), _p_p70S6k^T421/S424^ (**B**), and _p_mTOR^S2448^ (**C**) at rest (T0), 45 min after the 1st RE session (T1), after the 7th (T7) and 13th RE sessions (T13), and after 10 days of detraining (X-T14). Upper panel: Representative images of membranes probed for the respective proteins (phospho- and total protein). Lower panel: Box–whisker plots for the individual proteins: upper whisker (max), lower whisker (min), upper box (75th percentile), lower box (25th percentile), and median. _p_mTOR^S2448^, _p_p70S6k^T421/S424^, and _p_rpS6^S235/236^ were normalized to the total protein levels for each time point and individual subject. Normalization to the baseline (T0) was performed by dividing all time points of individual subjects by the total average of T0. Differently colored points represent data from individual subjects from the PR (gray dots) and CO (black dots) groups. * *p* < 0.05; ** *p <* 0.01; *** *p <* 0.001 with respect to T0; # *p* < 0.05 with respect to T1. (**D**) A Ponceau-S-stained PVDF membrane that was previously subjected to Western blotting and with samples of a representative subject showing all five time points.

**Figure 2 ijms-23-05431-f002:**
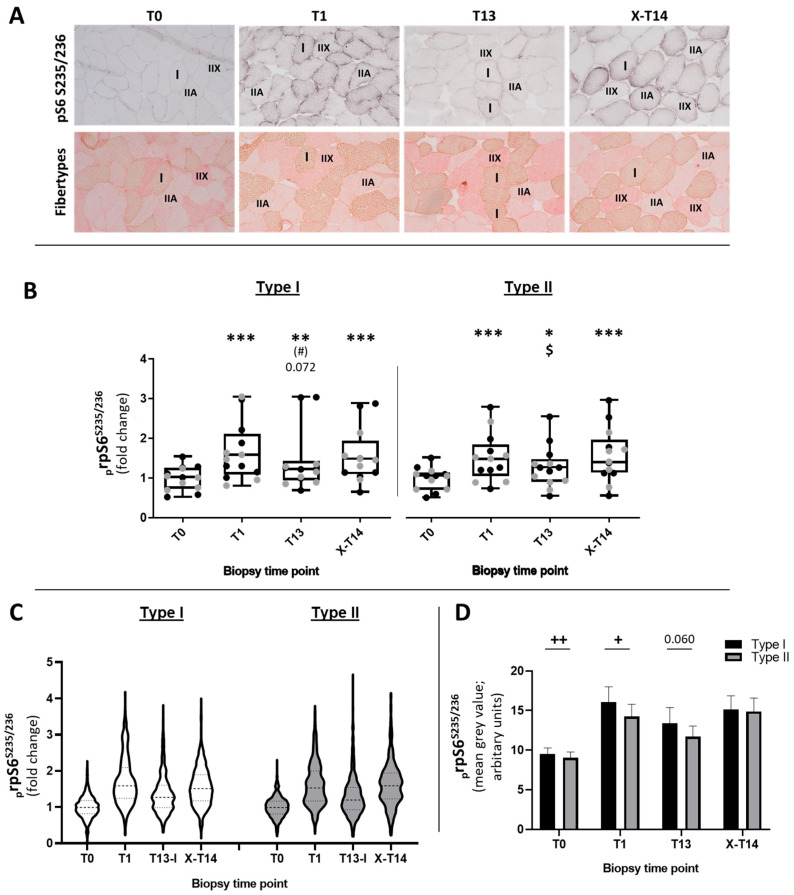
Quantification of the sarcoplasmic _p_rpS6^S235/236^ staining intensity in type I and II myofibers as determined via densitometry. (**A**) Representative images of muscle cross-sections (20-fold magnification) from one subject stained for _p_rpS6^S235/236^ (upper row of pictures) and the consecutive cross-section stained for type I (red–brown), type IIA (pale red), and IIX (full red) stained myofibers. (**B**) Box–whisker plots for all analyzed type I and II myofibers stained for _p_rpS6^S235/236^: upper whisker (max), lower whisker (min), upper box (75th percentile), lower box (25th percentile), and median. Fiber-type-specific _p_rpS6^S235/236^ data were normalized to the baseline (T0) by dividing all time points of individual subjects by the total average of T0. Differently colored points in the graphs represent data from individual subjects from the PR (gray dots) and CO (black dots) group. * *p* < 0.05; ** *p* < 0.01; *** *p* < 0.001 with respect to T0; # *p* < 0.05 with respect to T1; $ *p* < 0.05 with respect to X-T14 (**C**) Violin plots displaying the frequency of type I (white violins) and type II fibers (gray violins) with identical staining intensity of _p_rpS6^S235/236^ (horizontal width of the violin corpus) and the variable staining intensity of _p_rpS6^S235/236^ within the entire spectrum of analyzed fibers (*y*-axis). Normalization to the baseline was conducted by dividing the mean of all measured fibers of one subject and time point by the corresponding mean value of T0. (**D**) Direct comparison of _p_rpS6^S235/236^ levels between type I and type II fibers over the time course; + *p* < 0.05; ++ *p* < 0.01 for the difference between fiber types.

**Figure 3 ijms-23-05431-f003:**
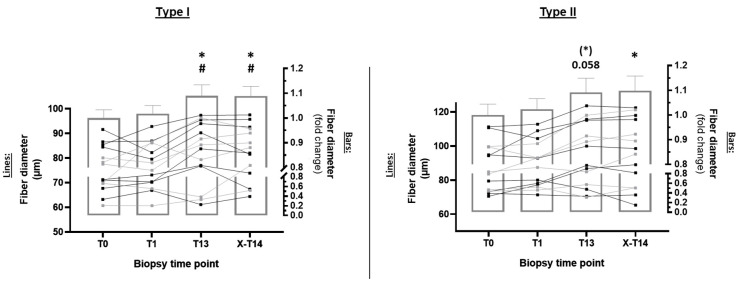
Determination of myofiber hypertrophy via analysis of type I and type II fiber diameter. Bar graphs display the mean ± standard error of the mean. The right *x*-axis represents the fold change in fiber diameter (T1, T13, and X-T14 compared to T0). Normalization to the baseline (T0) was performed by dividing all time points of individual subjects by the total average of T0. Lines (left *x*-axis) represent the absolute change in an individual subject’s fiber diameter (µm). Differently colored lines in the diagram represent subjects from CO (gray dots and lines) and PR (black dots and lines). * *p <* 0.05 with respect to T0; # *p* < 0.05 with respect to T1.

**Figure 4 ijms-23-05431-f004:**
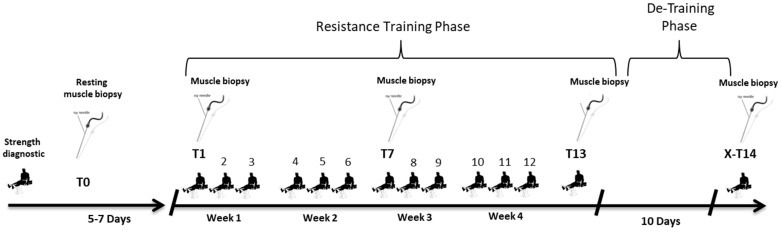
Study design. Subjects performed 14 training sessions in total within approximately six weeks. Three RE sessions per week were conducted on Mondays, Wednesdays, and Fridays until T13. Afterwards, a detraining period of 10 days without RE was conducted with a subsequent and final 14th RE session. Muscle biopsies were taken 5–7 days before the 1st (T0) and 45 min after the 1st (T1), 7th (T7), 13th (T13), and 14th (X-T14) training sessions.

## Data Availability

The data presented in this study are available on request from the corresponding author.

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
