# Peer review of "Repeated and Interrupted Resistance Exercise Induces the Desensitization and Re-Sensitization of mTOR-Related Signaling in Human Skeletal Muscle Fibers"

_ijms, 2022, doi:10.3390/ijms23105431_

Round 1

Reviewer 1 Report

Manuscript ijms-1704219 is interesting and analyzes how the repeated and interrupted resistance exercise induces the desensitization and re-sensitization of mTOR-related signaling in human skeletal muscle fibers.

The topic is certainly of interest and the manuscript was aimed at demonstrating how the activation / signaling of the mTOR-related signaling is modulated by the level of administration of the resistance exercise.

Unfortunately the data does not support this hypothesis, and it is really a shame.

All the limitations of the work, including references to works that already mentioned the desensitization / phosphorylation system of the mTOR-related signaling are extensively and elegantly described in the paper.

Surely one aspect that should be mentioned (but I understand the already important breadth of the manuscript) is a reference to muscle damage, muscle damage and muscle efficiency especially in relation to the presence of circulating markers of damage (CK LDH .

Another perplexing aspect is the effect of the protocol on the different types of fiber. I fear that due to the size of the sample and the individual variability some effect in the comparison between the different fibers is not very evident.

Author Response

We thank the reviewer for this comment. In line 489 we implemented a reference that highlights the relation between muscle damage and important circulating factors (CK and LDH). Of note, in the publication of Ulbricht et al., we also determined in our samples significantly increasing CK levels from T0 to T1 and a significant reduction already after T7. We linked this reference in line 491.

Another perplexing aspect is the effect of the protocol on the different types of fiber. I fear that due to the size of the sample and the individual variability some effect in the comparison between the different fibers is not very evident.

We thank the reviewer for mentioning this aspect. We determined the fiber type specific regulation on different perspectives in figure 3. From a methodological point of view, we have a high experience in standardizing all procedures to determine fiber type specific regulations on the protein levels via semiquantitative immunohistochemistry and published many papers on that issue. Of note, in figure 3 B and D the individual variability is clearly shown as individual dots alongside with the mean effect induced by training for each subject. Importantly, those data reflect (although weaker) also the western blotting results. We determined higher effect sizes for our data (ES was added in the results). However, the general comment is correct.  The effect is not a strong as in western blotting due to a larger variability. We analyzed only the sarcoplasmic parts of myofibers via densitometry whereas in western blotting the antibody recognizes all antigens in nuclei, sarcoplasm and basement membranes (See Figure 3A). Therefore, the fiber type specific analysis does not include all proteins of rpS6 e.g. That surely contributes to the difference between WB and immunohistochemistry.

However, we explicitly aimed to highlight the variability of a fiber specific regulation in our approach and displayed it in Figure 3 A and C. Only in figure C we descriptively show all analyzed fibers without a statistical approach to show that type I and II fibers do not exhibit a unique level of phosphorylation in muscle. This variability is surely also abundant in western blotting but cannot be shown with this method. We therefore believe that showing a fiber type specific regulation adds an important data set to the regular whole muscle western blotting procedure and its honest variability which is also the nature of human muscle data.

Reviewer 2 Report

This article seems well built and brings evidence of a physiological phenomenon not yet fully understood and that certainly deserves further study.

Some points of revision, suggested to the authors to increase the level of the paper, are provided below

Subjects

  • The authors need to include the sample size/power

Training intervention:

  • Unclear (specific reference) how the authors chosen the sets/reps/recovery in-between sets/reps  (https://pubmed.ncbi.nlm.nih.gov/24823345/) and normalized the external load 1-RM to better replicate this study
  • Unclear how the authors monitored the time under tension/speed on pushing phase on both machines

https://pubmed.ncbi.nlm.nih.gov/29050041/   https://pubmed.ncbi.nlm.nih.gov/30405437/), the setting for both machines, anyway in both machines the muscular activity is different, in-fact this methodological approach cannot clarify if these results are related to the leg press more than leg extension …..this point is very hard to overcome

  • I suggest changing terms about the Eccentric/concentric phase (https://pubmed.ncbi.nlm.nih.gov/24459543)

For each methodological approach and/or devices used, the authors need to include a specific reference about the validity/accuracy/precision

Statistical Analysis

  • Test retest is missing; therefore, the authors need to include this point in “Study Limitation”
  • I suggest to include the effect size (Cohen d)

Minor points

The authors should include “.” as decimal separator more than “,”

The figure 1 shown the leg extension….while the leg press in missing

Author Response

The authors need to include the sample size/power

We included this information in the methods part “subjects”

Training intervention:

Unclear (specific reference) how the authors chosen the sets/reps/recovery in-between sets/reps  (https://pubmed.ncbi.nlm.nih.gov/24823345/) and normalized the external load 1-RM to better replicate this study

Thank you for your comment: Our aim was to induce muscle growth, and a sufficient activation of signaling which is usually done with more repetitions (Burd et al. 2010 PMID: 20581041, Jacko et al 2019.). This is achieved at around 40% of 1 RM with 25 reps per set but also with 80% of 1 RM and 8 reps and importantly when conducting 5 sets and more (Terzis et al. 2010 PMID: 20617335; Jacko et al. 2019, Mitchell et al. 2012 PMID: 22518835). We initially thought about standardizing to the 1 RM but as the subjects had to train with 10-12 reps we chose to be closer to that load than the 1 RM. In our baseline strength diagnostics, we were thus able to precisely determine the dynamic forces along the required loading during RE training. Based on the work of other authors in the field and who usually control the training with equal methods we were also able to create a comprehensible loading. We did this on two different machines which are specific for the activation of the knee extensors over a time frame of 14 subsequent RE sessions.

Unclear how the authors monitored the time under tension/speed on pushing phase on both machines

We thank the reviewer for this important comment. We missed to include this information in the methods and implemented this information in the methods in lines 153-159.

https://pubmed.ncbi.nlm.nih.gov/29050041/   https://pubmed.ncbi.nlm.nih.gov/30405437/), the setting for both machines, anyway in both machines the muscular activity is different, in-fact this methodological approach cannot clarify if these results are related to the leg press more than leg extension …..this point is very hard to overcome

We thank the reviewer for this comment. The references provided by the reviewer refer to a comparison between leg press and squatting. We applied in our study leg extension followed by leg press in order to activate vastus lateralis muscle fibers during RE: Our aim was primarily to achieve a practical but comprehensible training situation which induces some degree of hypertrophy, significantly increases the phosphorylation of signaling proteins but also sufficiently recruits a comprising count of type I and II fibers (Jacko et al. 2019).  We achieved that aim based on the immunohistochemical data shown in figure 3.

Based on the literature, vastus lasteralis muscle is active during leg extension but also during leg press (Signorile et al. 1994) but the muscle activity is further affected by the speed of contraction (Padulo et al. 2017, Chalmers et al. 2008). We added some information highlighting the importance of considering this information for such RE-based studies.

By applying a defined combination of leg press and leg extension RE, combined with lengthening and shortening contractions and a standardized movement speed over the time-course of the study, we controlled all relevant parameters as good as possible. But we also mimicked a realistic training regimen. However, we acknowledge this important comment by the reviewer as it would be very interesting to determine the difference between leg press and leg extension concerning the molecular regulation of the mTOR pathway and especially mechanosensitive proteins and molecular damage markers. However, this was beyond the scope of the current approach. Interestingly, based on our previous works, where we analyzed only isokinetic knee extensions (Gehlert et al. 2015, Jacko et al. 2019), or in combination with leg extensions (Ulbricht et al. 2015, Jacko et al. 2021 and also here), there are basically similar results concerning the fiber type specific regulation of the molecular muscle machinery. The biggest difference concerning fiber type specific regulation seems to be the intensity in combination with the volume of loading.

I suggest changing terms about the Eccentric/concentric phase (https://pubmed.ncbi.nlm.nih.gov/24459543)

We thank the reviewer for this comment and added some additional information and the specific reference on this aspect in the methods section. This information was new for us and is obviously true. However, we feel that for many exercise scientists the nomenclature of eccentric and concentric is familiar and comprehensible. We therefore decided not to remove the description of concentric and eccentric. We also used this in the previous studies that were based on our samples (Jacko et al. 2019, 2020, Ulbricht et al. 2015, Gehlert et al. 2015). We hope that this additional information helps the reader to better differentiate this phenomenon.

For each methodological approach and/or devices used, the authors need to include a specific reference about the validity/accuracy/precision

The devices used for the biological analyses of the samples have a precision that is limited by the user (sample preparation etc.) and are also related in dependency of the protein content. This information does not make sense for all devices that work on a molecular level in this manuscript (microscopes, western blotting etc.).  However, the standardization of the movement speed was dependent on the movement sensor of the strength training machines. We added this information in the methods in line 155.

Statistical Analysis

Test retest is missing; therefore, the authors need to include this point in “Study Limitation”

We hope that we understand this comment correctly. Our study design was basically a single intervention with repeated measures and without a post test. Therefore, it was not necessary and also impossible to conduct a retest under these conditions. Interestingly, in a previous study from Jacko et al. 2019, we have determined, that the application of a second RE intervention on a preconditioned muscle (within 10 days after a first stimulation) significantly weakens the response also on molecular chaperones (HSPB5 phosphorylation). This emphasizes the nature of an adapting muscle and is likely the basis for our findings here.

I suggest to include the effect size (Cohen d)

We thank you for the intensive review of the data and your valuable comments. We included the effect size Kendalls W for data with the repeated measures Friedmann ANOVA and the effect size “r” for the fiber type specific analysis via the Wilcoxon test. We added the information in the statistics part and included the values at the respective sites in the results part and highlighted in red.

Minor points

The authors should include “.” as decimal separator more than “,”

We changed everything in the manuscript

The figure 1 shown the leg extension….while the leg press in missing

We thank you for the comment. However, the figure was just designed to illustrate the resistance exercise somehow graphically but not to specifically explain the entire training session

Reviewer 3 Report

Reviewer Report

In this study, the authors evaluated phosphorylation of mTOR-related anabolic signalling proteins in human skeletal muscle (musculus vastus lateralis) in response to resistance exercise (RE) training followed by a detraining phase of 10 days. Subjects performed a total of 14 training sessions. For the analysis were collected biopsies after each session. Data were generated by western blot as well as immunohistochemistry experiments. Initially, the authors aimed to investigated whether progression of RE intensity could prevent a drop in acute RE-induced phosphorylation compared to constant loading (CO). Since they did not observe a difference regarding the degree of hypertrophy and phosphorylation of mTOR, rpS6 and p70S6k between PR und CO, they included all subjects in one group.

This study provides novelties in molecular pathways, which are crucial for skeletal muscle functions.

Conclusions are appropriately supported by results.

However, however, some minor issues should be

resolved before publication.

  1. A general editing of the text is required
  1. The number of the ethic committee (line 522) should be mentioned
  2. The paragraph: tissue analysis, is missing (line 167)
  3. The “Author contributions” is missing (line 573)
  4. It is known that the internal ribosome entry site (IRES)-mediated translation initiation occurs independently of the 7-methyl guanosine cap at the 5′ end of the mRNA. This type of cap-independent translation provides an alternative to the canonical, cap-dependent mechanism of translation initiation (which depends on mTOR activation). By this alternative mechanism, myofibers might synthetize a subset of proteins under stress conditions or in response to prolonged resistance training. This important issue should be discussed about the observation that upregulation of protein synthesis occurs in a context in which mTOR-related anabolic signalling proteins phosphorylation is downregulated.  

Author Response

A general editing of the text is required

We edited the entire manuscript

The number of the ethic committee (line 522) should be mentioned

We added this information in the text

The paragraph: tissue analysis, is missing (line 167)

We added this information in the text

The “Author contributions” is missing (line 573)

We added the contributions.

It is known that the internal ribosome entry site (IRES)-mediated translation initiation occurs independently of the 7-methyl guanosine cap at the 5′ end of the mRNA. This type of cap-independent translation provides an alternative to the canonical, cap-dependent mechanism of translation initiation (which depends on mTOR activation). By this alternative mechanism, myofibers might synthetize a subset of proteins under stress conditions or in response to prolonged resistance training. This important issue should be discussed about the observation that upregulation of protein synthesis occurs in a context in which mTOR-related anabolic signalling proteins phosphorylation is downregulated. 

We appreciate this very worthful comment. We commented on the cap-dependent regulation in a previous paper, but unfortunately forgot to refer to this mechanism here. We implemented some sentences on that in the discussion (lines 522-532) and highlighted, that IRES may play an important role for the translation of crucial muscle specific proteins like ubiquitin ligases.

Round 2

Reviewer 2 Report

The main document was well improved. Congratulations